# Rice Seed Protrusion Quantitative Trait Loci Mapping through Genome-Wide Association Study

**DOI:** 10.3390/plants13010134

**Published:** 2024-01-03

**Authors:** Xiaowen Ding, Jubin Shi, Jinxin Gui, Huang Zhou, Yuntao Yan, Xiaoya Zhu, Binying Xie, Xionglun Liu, Jiwai He

**Affiliations:** College of Agronomy, Hunan Agricultural University, Changsha 410128, China; 18692271271@163.com (X.D.); shijubin10101@163.com (J.S.); 18390837980@163.com (J.G.); smileyyt1314@163.com (Y.Y.); zxy3339992021@163.com (X.Z.); 13550686697@163.com (B.X.)

**Keywords:** rice, protrusion percentage, QTL, GWAS, germplasm, haplotype

## Abstract

The germination of seeds is a prerequisite for crop production. Protrusion is important for seed germination, and visible radicle protrusion through seed covering layers is the second phase of the process of seed germination. Analyzing the mechanism of protrusion is important for the cultivation of rice varieties. In this study, 302 microcore germplasm populations were used for the GWAS of the protrusion percentage (PP). The frequency distribution of the PP at 48 h and 72 h is continuous, and six PP-associated QTLs were identified, but only *qPP2* was detected repeatedly two times. The candidate gene analysis showed that *LOC_Os02g57530 (ETR3)*, *LOC_Os01g57610 (GH3.1)* and *LOC_Os04g0425 (CTB2)* were the candidate genes for *qPP2*, *qPP1* and *qPP4*, respectively. The haplotype (Hap) analysis revealed that Hap1 of *ETR3*, Hap1 and 3 of *GH3.1* and Hap2 and 5 of *CTB2* are elite alleles for the PP. Further validation of the germination phenotype of these candidate genes showed that Hap1 of *ETR3* is a favorable allele for the germination percentage; Hap3 of *GH3.1* is an elite allele for seed germination; and Hap5 of *CTB2* is an elite allele for the PP, the germination percentage and the vigor index. The results of this study identified three putative candidate genes that provide valuable information for understanding the genetic control of seed protrusion in rice.

## 1. Introduction

As one of the three major cereals, rice (*Oryza sativa* L.) is an important food crop [1]. According to the Food and Agri culture Organization of the United Nations (FAO), the world’s rice-cultivated area is about 157 million hectares. China and India are the major rice producing countries. Between 1970 and 2019, global rice production increased from 379 million tons to 514 million tons. Seed germination is an important agronomic trait that affects crop yield and quality. The process of seed germination embraces three phases: imbibition, protrusion and germination [2,3,4,5]. During the initial imbibition phase of germination, dry seeds rapidly absorb water, and their quality increases. In the middle protrusion phase, the seeds reach a steady plateau phase with limited water absorption, and radicles break through the seed layers. The last phase is the seedling development phase [2,3,4,5]. Protrusion is the most important stage of seed germination, and physiologically, the arrival of the phase of protrusion is considered the completion of seed germination [3]. Protrusion is a complex physiological and biochemical process that involves metabolism reactivation; the mobilization of reserves; organelles, membranes and DNA repair; the synthesis of DNA, RNA and proteins; and coleoptile elongation [4,5]. Analyzing the mechanism of protrusion is very important for rice production [4]. To gain insight into the molecular mechanism of rice seed protrusion, we performed the seed protrusion percentage experiment to define the key factors involved in this process.

Seed germination is the process by which an embryo develops into a plumule and a radicle [2]. Enzymes such as amylase, protease and lipase, which separately solubilize starch, proteins and lipids, deliver glucose, amino acids and energy to a germinating embryo [2]. Seed germination is a complex agronomic trait controlled by many factors, such as seed dormancy, genotype, development, viability, storage time, hormones and the environment [6,7]. Numerous quantitative trait loci (QTLs) that are associated with seed germination have been reported in rice [8,9,10,11,12,13,14,15]. These QTLs have been identified through various experiments and studies, and they have been shown to play important roles in the regulation of seed germination. *qLTG3–1* is highly expressed in the embryo during seed germination and is associated with the weakening of tissues covering the embryo [8]. *qLTG3-1* affects a plant’s ability to germinate at low temperatures more easily [8]. He Y Q cloned *OsIPMS1* and found that the disruption of *OsIPMS1* resulted in low seed vigor under various conditions [16]. This reduction in seed vigor was affected by the decrease in amino acids during seed germination, including the amino acids associated with stress tolerance, GA biosynthesis and the TCA cycle. *OsIPMS1* could be used as a biomarker to determine the best time point for seed priming in rice [16]. *Sdr4* integrates the abscisic acid (ABA) and gibberellic acid (GA) signaling pathways at the transcriptional level and positively regulates seed dormancy by inhibiting active GA synthesis and promoting the accumulation of seed storage substances [17]. It is a central modulator of seed dormancy in rice. *OsGA20ox1* controls seedling vigor by being involved in GA biosynthesis [18]. Although studies on seed germination in rice have provided valuable insights, there is little research on seed protrusion. It is likely that there are many other genes and QTLs that play a role in seed protrusion that have not yet been identified.

Through a large-scale genomic data analysis, a GWAS can identify genotypes associated with rice QTLs such as yield, disease resistance and stress tolerance [19,20,21]. By performing a GWAS, Dong et al. [19] detected 30 QTL-controlled rice tillering angles from 529 rice accessions, including *TAC3*, *DWARF2* and *TAC1*. Chen et al. [20] detected five QTLs for rice grain shape by measuring the grain length and width of 289 different rice germplasms, including *GS3*, *GIF1*, *GW5* and *GSE9*. Li et al. [21] identified the rice blast resistance gene *bsr-d1* from rice Digu materials, which confers non-race-specific resistance to blast. Li et al. [22] identified a QTL, *qCTB4-1*, that is significantly correlated with drought resistance during the rice boot stage from 121 rice materials collected from the Mini Core Collection. A GWAS is performed using core accessions with strong polymorphisms in the target traits and has the advantage of studying quantitative traits in rice.

In this study, we conducted a GWAS for the protrusion percentage (PP) after the seed of accessions was imbibed for 48 h and 72 h, respectively, and detected QTLs to ascertain the seed protrusion percentage and preliminarily analyze candidate genes. Then, we selected 302 accessions from the 3K Rice Genome Project to confirm the haplotype and elite haplotype distribution of these candidate genes. Our results could contribute to understanding the genetic and molecular mechanisms of seed protrusion for the breeding of rice varieties.

## 2. Results

### 2.1. Phenotypic Variation in PP among 302 Accessions

There were large variations in the PP among the 302 accessions at 48 h and 72 h. The protrusion percentage ranged from 0% to 100% in 48 h and 72 h, with an average of 60.96% and 84.88%, and the coefficient of variation was 46.72% and 19.06%. The absolute values of the kurtosis were −0.06 and −2.28 and the skewness of the population was −0.83 and 6.47 (Appendix A), which indicates that the frequency distribution of the percentage of the protrusion showed a continuous distribution (Figure 1a,b). The statistical analysis of the protrusion percentage in the different subgroups in the two stages showed significant differences between Japonica and Indica (Figure 1c,d).

### 2.2. Population Structure of the 302 Rice Accessions

The genetic diversity and population structure of the 302 rice germplasm were analyzed using 198712 molecular markers. According to the principle of the maximum likelihood value, using ADMIXTURE 1.3.0 for the population structure analysis, it was found that when K = 11, CV error = 0.70814 was the smallest cross-validation error value. Therefore, the 302 varieties were divided into 11 subgroups (Figure 2c,d). The results of the cluster analysis (Figure 2a) and the principal component analysis (Figure 2b) are consistent with those of the population structure analysis.

### 2.3. GWAS for PP

The GWAS for the protrusion percentage (PP) was performed on the 302 accessions after seed imbibition for 48 h and 72 h. A total of six SNPs were identified as significantly associated with the PP based on the threshold *p*-value = 1.0 × 10^−5^. These SNPs were distributed on rice chromosomes 1, 2, 3, 4 and 9 (Table 1). Based on these significant SNPs, we finally identified six QTLs for the PP. Among these QTLs, *qPP2* was detected after seed imbibition for 48 h, *qPP1*, *qPP2*, *qPP2-1*, *qPP3*, *qPP4* and *qPP9* were detected after seed imbibition for 72 h (Figure 3), and only *qPP2* was repeatedly detected after the seed imbibition for 48 h and 72 h, indicating that *qPP2* was stably expressed at a different seed imbibition time.

### 2.4. Candidate Genes Identification for PP

In order to identify the six QTLs associated with the PP, we selected the genes mainly expressed in seeds on the basis of their expression profile in the Rice Expression Database (Appendix A) (http://expression.ic4r.org (accessed on 10 May 2023)) and, at the same time, removed the genes encoding retrotransposon or transposon proteins based on their functional annotations (https://www.rmbreeding.cn (accessed on 10 May 2023)). Seed germination is regulated by abscisic acid (ABA), gibberellins (GA), reactive oxygen species (ROS), reactive nitrogen species (RNS), auxin, cytokinin, ethylene, indoleacetic acid (IAA) and several other factors [1,2,3,4,5]. Among the 15 annotated genes, *LOC_Os01g58860* is involved in auxin signal transduction; *LOC_Os01g57610* participates in IAA synthesis; *LOC_Os02g57530* is involved in ethylene signaling; *LOC_Os01g59350*, *LOC_Os02g39810* and *LOC_Os02g57650* encode transcription factors; *LOC_Os01g57854* encodes a pectin esterase; *LOC_Os01g58750*, *LOC_Os02g39890*, *LOC_Os02g57180*, *LOC_Os03g26870* and *LOC_Os04g04254* are involved in endosperm development and are mainly expressed in seeds; *LOC_Os02g56850* encodes glutathione reductase and plays an important role in curtailing ROS; *LOC_Os03g26970* encodes an *α2* subunit of the 26S proteasome; and *LOC_Os09g11450* encodes a vacuolar *Na^+^*/*H^+^* antiporter. Detailed information on these 15 genes is listed in (Table 2).

### 2.5. Haplotype Analyses for PP Candidate Genes

The candidate genes were then identified using high-density association and gene-based haplotype analyses. Finally, three candidate genes for *qPP1*, *qPP2* and *qPP4* were obtained. No suitable candidate genes were found for *qPP2-1*, *qPP3* and *qPP9* based on the results of the haplotype analyses. Among the 15 novel candidate genes, the ethylene receptor (ETR) gene *LOC_Os02g57530 (ETR3)*, the indole-3-acetic acid–amido synthetase gene *LOC_Os01g57610 (GH3.1)* and the UDP-glucose sterol glucosyltransferase *LOC_Os04g04254 (CTB2)* were selected for further analysis.

#### 2.5.1. Haplotype Analyses for *LOC_Os02g57530 (ETR3)*

The annotated gene with the most significant hit was *ETR3* (Figure 4a). A previous report showed that *OsETR2*, *OsETR3* and *OsETR4* exhibited significant homology to the prokaryotic two-component signal transducer and a wide range of ethylene receptors [48]. The α-amylase gene *RAmy3D* was suppressed in *ETR2*-overexpressing plants but enhanced in the *etr2* mutant [49]. Under the conditions of ethylene-induced germination, the coleoptile growth of *etr2* and *etr3* was promoted [49]. The LD heatmap showed a moderate LD level around the *ETR3* gene (Figure 4a). Two major haplotypes were detected among the 302 accessions based on three SNPs in the *ETR3* 5′-UTR region, four SNPs in the coding region and three SNPs in the 3′-UTR regions (Figure 4b). The PPs of the varieties containing *ETR3* Hap3 were significantly lower than those of *ETR3* Hap1, and Hap1 had the highest mean PP (90.8%) (Figure 4c,d). Hap1 was mainly composed of the XI-adm subgroups, and Hap3 was mainly composed of the GJ-tmp and GJ-trp subgroups (Figure 4c). A significant difference in the germination percentage was observed among the Hap1 and Hap3 haplotypes (Figure 4e). Hap1 had the highest mean germination percentage (96.3%) (Figure 4e). Significant differences for the vigor index and shoot length were observed among the Hap2 and Hap3 haplotypes (Figure 4f,g). Hap2 showed the highest mean vigor index (2213.3) (Figure 4f) and root length (12.6 cm) (Figure 4h). Hap2 is a favorable allele for the vigor index and root length of *ETR3*.

#### 2.5.2. Haplotype Analyses for *LOC_Os01g57610 (GH3.1)*

In the region of *qPP1*, the annotated gene was *GH3.1* (Figure 5a). *GH3* is one kind of early auxin-responsive gene that widely exists in numerous plants [44]. Indoleacetic acid (IAA) was shown to be involved in the early stages of seed germination in many species [50]. The indole-3-acetic acid (IAA)–amido synthetase gene GRETCHEN HA-GEN3-2 (*OsGH3-2*) is associated with seed storability, contributing to the wide variation in seed viability between the populations after long periods of storage and artificial ageing [51]. *OsGH3.1* is an indole-3-acetic acid (IAA) amido synthetase, whose homolog in Arabidopsis functions in maintaining auxin homeostasis by conjugating excess IAA to various amino acids [52]. To investigate the causative SNP variations in *GH3.1* responsible for the phenotypic variations in the PP, we analyzed the SNPs in the genomic coding region of *GH3.1* across the 302 varieties, which revealed four major haplotypes (Figure 5b). These four major haplotypes were based on two SNPs in the *GH3.1* 5′-UTR region, three SNPs in the coding region and five SNPs in the 3′-UTR region (Figure 5b). Significant differences for the PP were observed among the four haplotypes except between Hap1 and Hap2 and Hap4. Hap1 had the highest mean PP (91.6%) and showed a significantly higher mean PP than Hap2 and Hap4 (Figure 5c). The mean PP of Hap3 was 89.6%. Hap1 was mainly composed of the XI-adm subgroup, Hap2 was mainly composed of the GJ-trp subgroup, Hap3 was mainly composed of the XI-2 subgroup, and Hap4 was mainly composed of the GJ-tmp subgroup (Figure 5d). The haplotype (Hap) analysis revealed that Hap1 and 3 of *GH3.1* are favorable alleles for the PP. We furthermore detected the germination percentage, vigor index, shoot length and root length in accessions with different haplotypes. Hap3 had the highest mean germination percentage (96.7%) (Figure 5e), vigor index (2519.4) (Figure 5f), shoot length (6.1 cm) (Figure 5g) and root length (13.3 cm) (Figure 5h) in accessions with different haplotypes. Hap3 of *GH3.1* is a favorable allele for seed germination.

#### 2.5.3. Haplotype Analyses for *LOC_Os04g04254 (CTB2)*

In the region of *qPP4*, the annotated gene was *CTB2*(Figure 6a,b). The *CTB2* gene, which encodes a UDP-glucose sterol glucosyltransferase, is responsible for cold tolerance in rice at the booting stage [35]. To understand how the *CTB2* sequence may affect the PP phenotype, we analyzed the SNPs in the genomic coding region of *CTB2* across the 302 varieties, which revealed five major haplotypes, two SNPs in the coding region and seven SNPs in the intron region (Figure 6b). Hap1, Hap2, Hap3, Hap4 and Hap5 contain 17, 22, 19, 72 and 63 accessions, respectively. Significant differences for the PP were observed between Hap5 and Hap1, Hap3 and Hap4. Hap5 had the highest mean PP (92.1%) and showed a significantly higher mean PP than Hap1 and Hap3 (Figure 6b,c). Hap2 had the second-highest mean PP (88.5%). Hap1 and Hap3 were mainly composed of the GJ-tmp subgroups, and Hap5 was mainly composed of the XI-1A and XI-adm subgroups (Figure 6d). Hap5 had the highest mean germination percentage (96.1%). Accessions with Hap5 had the highest mean vigor index (2415.2) (Figure 6e,f). There were no differences between the Haps in terms of shoot length or root length (Figure 6g,h). The haplotype (Hap) analysis revealed that Hap2 and Hap5 of *CTB2* are favorable alleles for the PP, germination percentage and vigor index; Hap5 is a favorable allele for the vigor index. The results showed that Hap5 of *CTB2* is a favorable allele for seed germination.

## 3. Discussion

Seed germination is a complex quantitative trait controlled by many genetic factors in the embryo, aleurone layer, endosperm and pericarp [2,3,4,5]. The germination process is an initial and important step in the production of agricultural products. Rapid seedling establishment is an important agronomic trait for direct seedling in rice [53]. Seed imbibition is the first step of seed germination [4]. Protrusion is the second stage of seed germination, during which the activation and repair of biological macromolecules and organelles occurs, the seed embryo cells resume growth, and the tip of the radicle breaks through the seed coat [4]. Physiologically, the arrival of the phase of protrusion is considered the completion of seed germination. Previous studies about seed germination have conducted linkage analyses and association analyses by using different genetic populations and natural populations to construct genetic linkage maps [11,29,30,31]. QTLs/genes controlling seed germination have been identified by using genetic, molecular biology and biochemical methods, and candidate genes for germination have been screened and functionally validated to explore the molecular mechanisms of seed germination regulation [8,9,10,11,12,13,14,15,16,17,18]. A GWAS is a powerful approach to determining genes associated with seed germination in rice. *qSP3* for the seedling percentage was identified, and *OsCDP3.10* is the *qSP3* candidate gene that regulates seed vigor and is involved in the ROS level [53]. *qSRMP9* for rice seed reserve mobilization was validated, and cytochrome b5 (*OsCyb5*) is the *qSRMP9* candidate gene that regulates seed reserve mobilization and seedling growth [54]. Under salt stress, 11 loci associated with seed vigor were detected, and two candidate genes, *OsNRT2.1* and *OsNRT2.2,* encoding nitrate transporters, were identified [55]. Currently, there are some related reports on the molecular mechanisms regulating rice germination, but the molecular regulation network mechanism controlling rice seed protrusion has not been explored in depth.

### 3.1. Abundant Variation and GWAS Results of PP in Rice Germplasm

Seed protrusion is the most important trait in rice seedling growth. In this study, for 302 rice germplasm populations, the protrusion percentage phenotype was identified at 48 h and 72 h after imbibition, and the results of the phenotype identification, carried out three times, showed that the protrusion percentage phenotype of the germplasm population varied widely. The statistical analysis of the protrusion percentage in the different subgroups in the two stages showed significant differences between Japonica and Indica (Figure 1d). In our study, six PP-associated QTLs were detected after 48 h and 72 h, and only *qPP2* was repeatedly detected two times.

### 3.2. Identification of Candidate Genes for PP

The endogenous plant hormones ABA, GA, ethylene and IAA have been reported to affect seed germination [2]. *Sdr4* positively regulates seed dormancy by inhibiting active GA synthesis, and *OsGA20ox1* is involved in gibberellin (GA) biosynthesis, which is important for seed germination [17,18]. *OsTPP1* controls seed germination through crosstalk with the ABA catabolic pathway [56]. In this study, by consulting the relevant literature, the expression profiles in the Rice Expression Database and the results of haplotype analyses, *ETR3* was finally identified as a candidate gene for *qPP2*. It encodes an ethylene receptor, three SNPs in the *ETR3* 5′-UTR region, four SNPs in the coding region and three *SNPs* in the 3′-UTR region, causing significant differences in the PP among the three haplotypes. Hap2 of *ETR3* is a favorable allele for the PP and germination. *GH3.1*, an in-dole-3-acetic acid (IAA)–amido synthetase, has two SNPs in the 5′-UTR region, three SNPs in the coding region and five SNPs in the 3′-UTR region. Hap3 of *GH3.1* is a favorable allele for the PP and germination. *CTB2*, which encodes a UDP-glucose sterol glucosyltransferase, has two SNPs in the coding region and seven SNPs in the intron region, and Hap 5 of *CTB2* is a favorable allele for the PP.

## 4. Materials and Methods

### 4.1. Plant Materials

We used 302 germplasm resources from the 3K Rice Genome Project from 36 countries, including China (67), India (51), Bangladesh (22), the Philippines (29), etc. These germplasm resources were divided into 12 subgroups: 13 in admix, 20 in cA (Aus), 7 in cB (Bas), 7 in GJ-adm, 8 in GJ-sbtrp, 29 in GJ-tmp, 32 in GJ-trp, 31 in XI-1A, 24 in XI-1B, 36 in XI-2, 34 in XI-3 and 61 in XI-adm [56]. Detailed information regarding these varieties is listed in Appendix A. The 302 germplasm resources were planted in Ling shui Xian, Hainan Province, in 2022. The seeds were sown on 22 November, and the seedlings were transplanted from 20 to 21 December. Normal field production was used for field management. Rice seeds were transferred to nursery beds for germination after being soaked in water at 30 °C for 48 h. Seedlings that were 20 days old were then transplanted to the paddy field. Accessions had six plants per row, one seedling per hill, with a density of 20 × 20 cm, and each accession was planted in 4 rows. The seeds of each accession were harvested individually after maturity. The harvested seeds were dried in a hot air dryer at 37 °C for 5 days and then stored at room temperature for three months [35].

### 4.2. Phenotypic Identification of PP

One hundred plump seeds were placed in sterile petri dishes that contained distilled water and kept at 30 °C in a growth chamber with a 24 h dark photoperiod to facilitate protrusion in the dark [36]. The protrusion percentage (the protrusion criterion is based on the radical length = 1 mm) was scored after imbibition for 48 and 72 h. Three replications were carried out on each plate containing 100 seeds.

### 4.3. Phenotypic Identification of Seed Germination

One hundred plump seeds from each accession were placed in petri dishes (d = 12 cm) with two sheets of filter paper and 20 mL of sterile distilled water added. These seeds were then incubated in a growth chamber at 30 °C for 7 days with a 12 h light/12 h dark photoperiod to promote seedling growth. After 7 days of incubation, the germination percentage (GP) of the seeds in each dish was calculated, and 10 flax seedlings were randomly selected to measure the shoot length (SL) and root length (RL). The vigor index (VI) of the seedlings was determined: *V I* = *(RL* + *SL)* × *GP*.

### 4.4. Population Structure Analysis

Highly redundant SNPs were removed by pruning the LD-based SNPs (r2 > 0.3) using PLINK 1.9. After PLINK pruning, 198712 SNPs were used to validate the subpopulation classification and origin of the 302 rice germplasms. Population structure analysis was performed using Admixture 1.3 software [57]. The number of ancestral populations (K) was assumed to be between 2 and 6. Principal component analysis (PCA) was performed using the PLINK 1.9 software based on the R’ggplot2′ package [58]. Phylogenetic analyses were performed with Fast Tree 2.1 software using the approximate maximum likelihood method [59].

### 4.5. Genome-Wide Association Study Analysis

We conducted two separate GWAS analyses using phenotypic data from all 302 materials that measured the protrusion percentage (PP) after the imbibition of germplasm accessions’ seeds for 48 h and 72 h. The raw genotype data of the 302 accessions were obtained from the Rice Diversity Database (http://www.ricediversity.org (accessed on 6 Jun 2023)). After discarding the heterozygous markers and those with missing data >20% and with minor allele frequency <5%, the remaining 198,712 SNP markers were used for Genome-Wide Association Studies (GWASs). Using population structure (Q matrix) and kinship relatedness data (K matrix), we used the mixed linear statistical model (MLM) to perform an association analysis between the phenotypic traits and the SNP data [38]. The threshold *p*-value was set at 1.0 × 10^−5^, and the SNP with the smallest *p*-value in the cluster was considered the leading SNP. The QTL interval was defined as the 100 kb region on both sides of the QTL peak position, and the highest R2 value represented the contribution rate of the corresponding association region [60]. Manhattan and quantile–quantile (Q–Q) plots were generated by using the R package ‘CMplot’.

### 4.6. Candidate Gene Screening and Linkage Disequilibrium (LD) Analysis

Candidate genes in the QTL intervals were extracted from the database website (https://www.rmbreeding.cn (accessed on 10 May 2023), and gene-based haplotype analyses using 302 accessions from the 3K RGB germplasm were carried out to detect candidate genes of QTLs for protrusion. High-quality SNPs were used for the analysis, and the gene with the most significant hit within a local LD block constructed around the QTLs was screened as the candidate gene [61,62]. The R package ‘LDheatmap’ was used to draw the heatmap of pairwise LDs.

### 4.7. Haplotype Analysis

The haplotype for each candidate gene was created by concatenating the SNPs within 2 kb of the upstream initiation codon (promoter regions), 3′ and 5′ untranslated regions (UTR) and nonsynonymous SNPs in the coding regions. Multiple comparisons used haplotypes carried by at least 15 accessions [60].

### 4.8. Statistical Analysis

The data analysis was performed using SPSS 25.0 (SPSS Inc., Chicago, IL, USA), and the results are expressed as the mean values ± SD. The statistical assessment of the data was analyzed with Duncan’s multiple comparison test (at a 5% significance level) following a one-way ANOVA.

## 5. Conclusions

In this study, we conducted genome-wide association mapping for seed protrusion based on high-density SNPs using 302 rice accessions. We identified six PP-associated QTLs and screened three candidate genes: *LOC_Os02g57530 (ETR3)*, *LOC_Os01g57610 (GH3.1)* and *LOC_Os04g0425 (CTB2)*. Our study provides new insights into the genetic basis of seed protrusion in rice. The identification of these candidate genes and their elite haplotypes could be useful for rice production and will be a promising source for the molecular breeding of ideotypes in rice.

## Figures and Tables

**Figure 1 plants-13-00134-f001:**
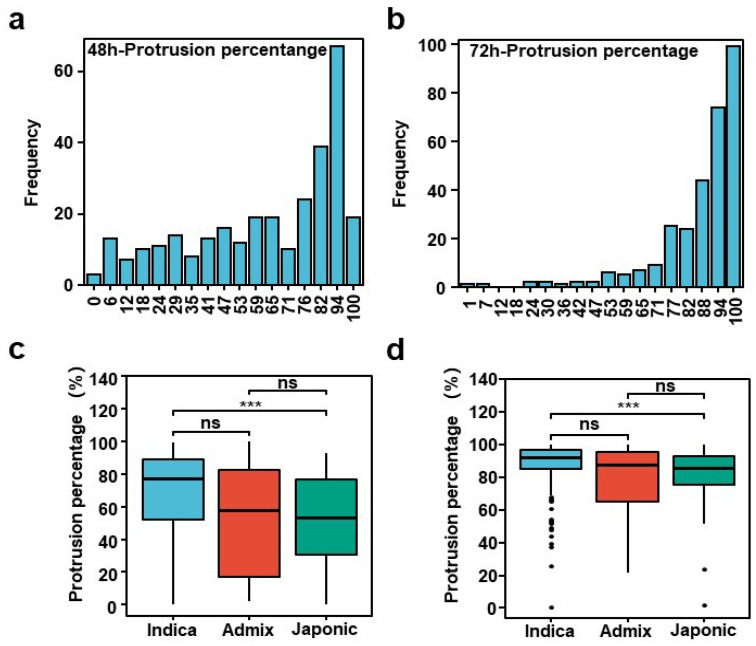
The variation in PP in 302 rice accessions; (**a**,**b**) distribution of PP in 302 accessions after imbibition for 48 h and 72 h, respectively; (**c**,**d**) multiple comparisons of protrusion percentage in different subgroups after imbibition for 48 h and 72 h. Data represent mean ± SD of three replicates. *** *p* ≤ 0.001; ns: not significant.

**Figure 2 plants-13-00134-f002:**
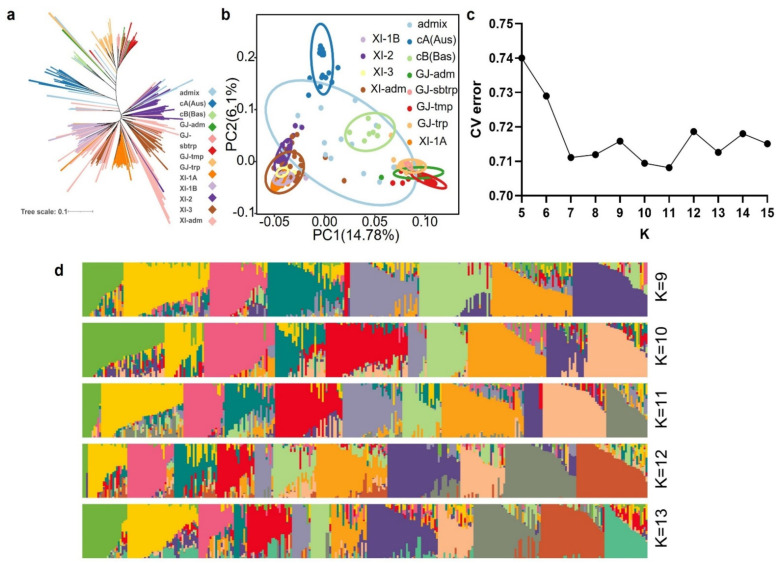
Population structure and phylogenetic analysis of 302 rice accessions. (**a**) Phylogenetic trees of 302 rice accessions. (**b**) PCA plots for the 302 rice accessions. (**c**) Cross-validation error of K value. (**d**) ADMIXTURE analyses for k = 9–13.

**Figure 3 plants-13-00134-f003:**
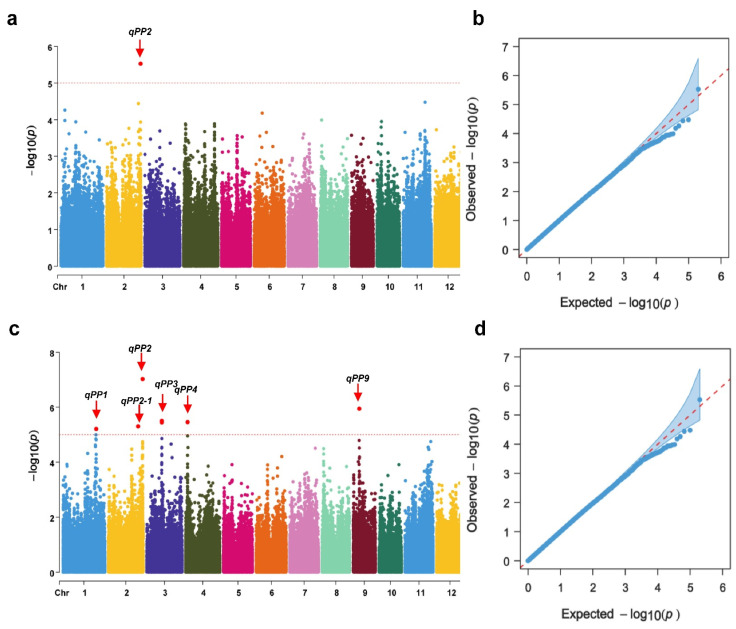
GWAS for protrusion percentage in rice. (**a**,**c**) Manhattan plots of GWAS after the germplasm accessions imbibition for 48 h and 72 h, respectively; (**b**,**d**) Q–Q plots of GWAS after the germplasm accessions imbibition for 48 h and 72 h, respectively.

**Figure 4 plants-13-00134-f004:**
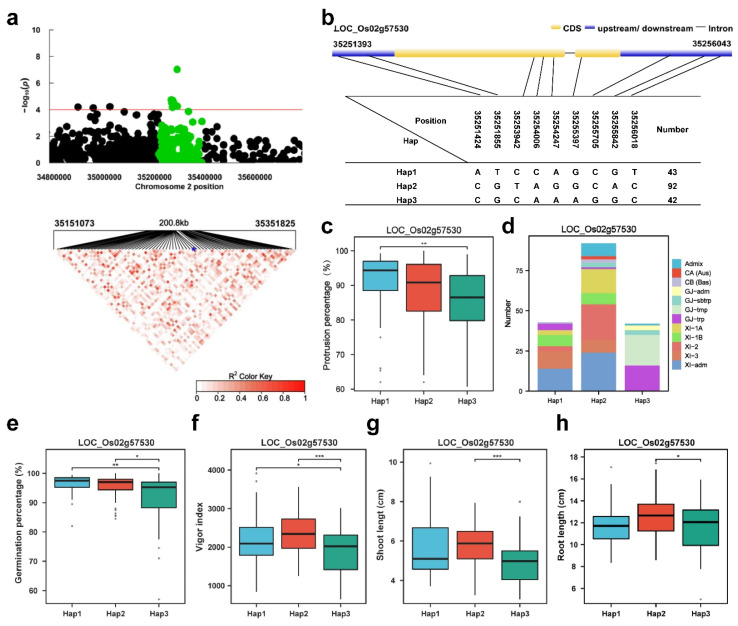
*LOC_Os02g57530 (ETR3)* haplotype significance analysis. (**a**) Plot of linkage disequilibrium for SNPs with -log10 *p*-value > 4 in *qPP2* on Chr.2; (**b**) diagram of *ETR3* structure and the positions of 7 SNPs used for haplotype analysis, bar = 100 bp; (**c**) comparison of the PP values between accessions and different haplotypes. (**d**) subpopulation composition of *ETR3* haplotypes for PP; (**e**–**h**) comparisons of germination percentage (%), vigor index, shoot length (cm) and root length (cm) among accessions with different haplotypes. Blue asterisk: Position of *LOC Os02g57530* (*ETR3*). * *p* ≤ 0.05; ** *p* ≤ 0.01; *** *p* ≤ 0.001.

**Figure 5 plants-13-00134-f005:**
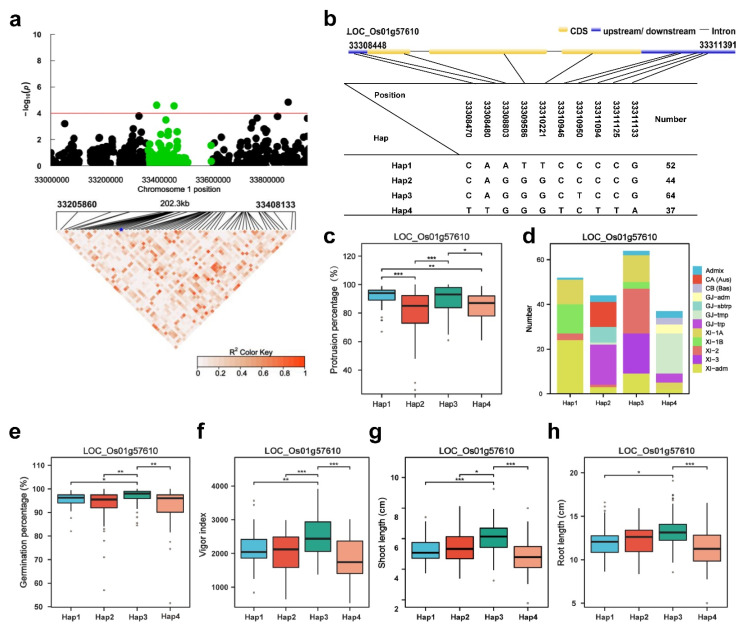
*LOC_Os01g57610 (GH3.1)* haplotype significance analysis. (**a**) Plot of linkage disequilibrium for SNPs with −log10 *p*-value > 4 in *qPP1* on Chr.1; (**b**) diagram of *GH3.1* structure and the positions of 7 SNPs used for haplotype analysis, bar = 100 bp; (**c**) comparison of the PP values between accessions and different haplotypes. (**d**) subpopulation composition of *GH3.1* haplotypes for PP; (**e**–**h**) comparisons of germination percentage (%), vigor index, shoot length (cm) and root length (cm) among accessions with different haplotypes. Blue asterisk: Position of *LOC Os01g57610* (*GH3.1*).* *p* ≤ 0.05; ** *p* ≤ 0.01; *** *p* ≤ 0.001.

**Figure 6 plants-13-00134-f006:**
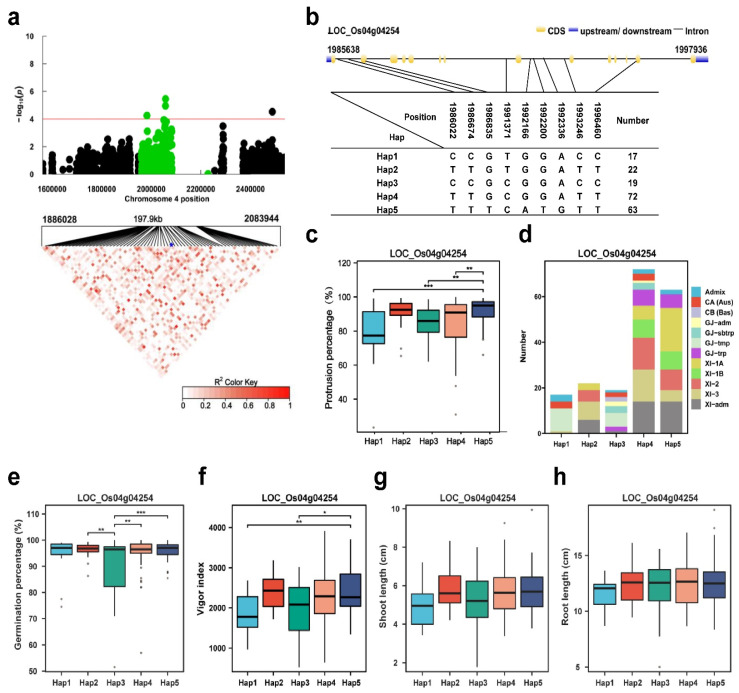
*LOC_Os04g04254 (CTB2)* haplotype significance analysis. (**a**) Plot of linkage disequilibrium for SNPs with −log10 *p*-value > 4 in *qPP4* on Chr.4; (**b**) diagram of *CTB2* structure and the positions of 7 SNPs used for haplotype analysis, bar = 100 bp; (**c**) comparison of the PP values between accessions and different haplotypes. * *p* ≤ 0.05; ** *p* ≤ 0.01; *** *p* ≤ 0.001; (**d**) subpopulation composition of *CTB2* haplotypes for PP; (**e**–**h**) comparisons of germination percentage (%), vigor index, shoot length (cm) and root length (cm) among accessions with different haplotypes. Blue asterisk: Position of *LOC Os04g04254* (*CTB2*). * *p* ≤ 0.05; ** *p* ≤ 0.01; *** *p* ≤ 0.001.

**Table 1 plants-13-00134-t001:** List of QTLs for protrusion percentage, identified by GWAS.

Time	QTL	Chr	Position	*p*-Value	Effect	Known QTL
48 h	*qPP2*	2	35289602	2.97 × 10^−6^	−9.172954	
72 h	*qPP1*	1	34261659	6.21 × 10^−6^	10.989112	*qNaK1.11, qRTL1.26, qSRR1.29* [23,24,25]; *qDWT1.21* [26]; *qAN4d-S1* [27]
	*qPP2-1*	2	24026046	8.42 × 10^−5^	−6.885124	*qCSH2* [28]
	*qPP2*	2	35289602	9.51 × 10^−8^	−6.040726	*qAG-2-8* [29]
	*qPP3*	3	15508309	3.17 × 10^−6^	8.9194262	*qGR3.2, qDOR-3-1* [30]; *qDOM3.4* [31]; *qAG3* [11]
	*qPP4*	4	2057628	3.51 × 10^−6^	7.1252147	*qCBT4-1* [22,32,33]
	*qPP9*	9	6224429	1.14 × 10^−6^	7.3173322	*qLTGR4d-9-1* [34]; *qAG9-1* [29]

**Table 2 plants-13-00134-t002:** Candidate genes and function annotations of QTLs for PP.

QTL	Candidate Gene	Description	Reference
*qPP1*	*LOC_Os01g57610*	*GH3.1*: indole-3-acetic acid–amido synthetase gene	*GH3.1* [35]
	*LOC_Os01g57854*	*OsPME1*: pectin esterase	*OsPME1* [36]
	*LOC_Os01g58750*	*OsGCD1:* gamete cells defective1	*OsGCD1* [37]
	*LOC_Os01g58860*	*OsPIN9*: auxin efflux carrier domain-containing protein	*OsPIN9* [38]
	*LOC_Os01g59350*	*OsbZIP08*: BZIP transcription factor	*OsbZIP08* [39]
*qPP2-1*	*LOC_Os02g39810*	Zinc finger: PHD-type domain-containing protein	
	*LOC_Os02g39890*	*du3*: dull endosperm 3	*du3* [40]
*qPP2*	*LOC_Os02g56850*	*OsGR2*: glutathione reductase	*OsGR2* [41]
	*LOC_Os02g57180*	*FLO13*: floury endosperm	*FLO13* [42]
	*LOC_Os02g57650*	*OsNAC78*: NAC (NAM, ATAF and CUC) transcription factor	*OsNAC78* [43]
	*LOC_Os02g57530*	*ETR3*: ethylene receptor	*ETR3* [44]
*qPP3*	*LOC_Os03g26870*	*SRWD5*: WD40 subfamily protein	*SRWD5* [45]
	*LOC_Os03g26970*	*OsTT1*: thermo-tolerance 1	*OsTT1* [46]
*qPP4*	*LOC_Os04g04254*	*CTB2*: cold tolerance at booting stage 2	*CTB2* [22]
*qPP9*	*LOC_Os09g11450*	*OsNHX1:* vacuolar Na^+^/H^+^ antiporter gene	*OsNHX1* [47]

## Data Availability

The data will be available upon specific request to the authors.

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
