# Peer review of "Rice Seed Protrusion Quantitative Trait Loci Mapping through Genome-Wide Association Study"

_plants, 2024, doi:10.3390/plants13010134_

Round 1
Reviewer 1 Report
Comments and Suggestions for Authors
The manuscript deals with the a GWAS approch to detect candidate gene involved in protrusion in a large collection of rice accessions.
The topic is interesting and involves plant sciences approaches meeting the expectations of Plants
The used molecular methods are current and wer performed soundly for both sampling and measurements. statistical analyses are almost convenient.
Nevertheless, due to the lack of literature background there are many confusing concepts and insufficient expertise in physiological stages of germination.
There are, therefore two major shortcomings
1- the stage of protrusion is very limited in duration. Moreover, it has been not defined which will probably lead readers to confuse this protrusion with germination. the authors need to define precisely this specific stage of germination and the limits (time, seed weight, imbibition rate...) of its appearance, using already published studies. This is crucial, since the authors have chosen two imbibition durations in their work that are very close and may overlap with this protrusion phase.
2-limits are not clearly defined for this phase which is shown in figure 1 plot C. The curve is typical of the germination with two phases. The first one is constant increase typical of imbibition and the second part (a a steady plateau) with limited water absoption. Unfortunately authors did not use the right equation of the curve. this distinction between the two parts of the curve is necessary to define the protrusion phase. Its seems confusing here in ths presentation. Please modify and explain.
Other remarks,
For Plants, M&M section should be placed after discussion and before conclusion. Please modify.
Author Response
For research article
Title: Rice Seed Protrusion QTL Mapping Through Genome-wide Association Study
Authors: Xiaowen Ding1, Jubin Shi1, Jinxin Gui 1, Huang Zhou 1, Yuntao Yan 1, Xiaoya Zhu 1, Binying Xie 1, Xionglun Liu 1,∗and Jiwai He 1,∗
|
Response to Reviewer 1 Comments
|
||
|
1. Summary |
|
|
|
Thank you very much for taking the time to review this manuscript. We appreciate the time and efforts that you dedicated to providing feedback on our manuscript and are grateful for the insightful comments on and valuable improvements to our paper. We have incorporated most of the suggestions. Those changes are highlighted within the manuscript. Please see below, in red, for a point-by-point response to the comments and concerns. Please find the detailed responses below and the corresponding revisions in the resubmitted files.
|
||
|
Comments 1: 1-the stage of protrusion is very limited in duration. Moreover, it has been not defined which will probably lead readers to confuse this protrusion with germination. the authors need to define precisely this specific stage of germination and the limits (time, seed weight, imbibition rate...) of its appearance, using already published studies. This is crucial, since the authors have chosen two imbibition durations in their work that are very close and may overlap with this protrusion phase. |
|
Response 1: Thank you for pointing this out. I agree with this comment. Therefore, we have added a paragraph as follows: During the initial imbibition phase of germination, dry seeds rapidly absorb water and quality increases. The middle protrusion phase, the steady plateau phase with limited water absorption and radicle breaks through the seed layers. The last phase is the seedling development phase [2–5].
|
|
Comments 2: 2-limits are not clearly defined for this phase which is shown in figure 1 plot C. The curve is typical of the germination with two phases. The first one is constant increase typical of imbibition and the second part (a a steady plateau) with limited water absorption. Unfortunately, authors did not use the right equation of the curve. this distinction between the two parts of the curve is necessary to define the protrusion phase. Its seems confusing here in ths presentation. Please modify and explain. |
|
Response 2: Thank you for pointing this out. We agree with this comment. The process of seed germination embraces three phases: imbibition, protrusion, germination. The first one is constant increase typical of imbibition and the second part (a steady plateau) with limited water absorption. In our opinion, the relationship between the Protrusion percentage of rice seeds after imbibition 48h and 72h should be causal. Therefore, it is incorrect to use linear correlation analysis. Given the above considerations, we have decided to exclude the Figure 1 plot C. Firstly, we believe that this result is not very important and deleting it does not affect the structure of this article. Secondly, we did not find a more appropriate model, so we finally decided to delete it. Comments 3: For Plants, M&M section should be placed after discussion and before conclusion. Please modify. Response 3: Thank you for pointing this out. I have placed the M&M section after the discussion and before the conclusion.
|
Many grammatical or typographical errors have been revised. All the lines and pages indicated above are in the revised manuscript.
Thank you for the kind advice. Merry Christmas and happy New Year!
Sincerely yours,
Xiaowen Ding
Reviewer 2 Report
Comments and Suggestions for Authors
Dear Authors,
Thank you for your submission. Nice work. I have some questions and suggestions and would like you to answer and include it in the manuscript before further considerations.
1- The title needs to be changed to reflect the work in a better way. Utility station of Genome wide……
2- Line 17-19 - add details of rice production, acerage, major producers etc.
3- Add a paragraph on GWAS and earlier studies in rice where it has been used to detect important QTLs.
4- Line 52- Line 119- What was the basis of selection of 302 germplasm lines?
5- Line 66- Specify standard field production techniques.
6- Line 74- Do you have image to show how this was done?
7- Whereis the graph to show peak for K analysis for STRUCTURE?
8- Include the results for STRUCTURE analysis before moving to GWAS.
Comments on the Quality of English Language
Quality of English language is good. Minor corrections needed.
Author Response
For research article
Title: Rice Seed Protrusion QTL Mapping Through Genome-wide Association Study
Authors: Xiaowen Ding1, Jubin Shi1, Jinxin Gui 1, Huang Zhou 1, Yuntao Yan 1, Xiaoya Zhu 1, Binying Xie 1, Xionglun Liu 1,∗and Jiwai He 1,∗
|
Response to Reviewer 2 Comments
|
||
|
1. Summary |
|
|
|
Thank you very much for taking the time to review this manuscript. We appreciate the time and efforts that you dedicated to providing feedback on our manuscript and are grateful for the insightful comments on and valuable improvements to our paper. We have incorporated most of the suggestions. Those changes are highlighted within the manuscript. Please see below, in red, for a point-by-point response to the comments and concerns. Please find the detailed responses below and the corresponding revisions in the resubmitted files.
|
||
|
Comments 1: The title needs to be changed to reflect the work in a better way. Utility station of Genome wide…… |
|
Response 1: Thank you for pointing this out. We agree with this comment. Therefore, we have changed title to Rice Seed Protrusion QTL Mapping Through Genome-wide Association Study.
|
|
Comments 2: Line 17-19 - add details of rice production, acreage, major producers etc. |
|
Response 2: Agree. I have added details of rice production, acreage, major producers at line17-19. As follows: According to the Food and Agriculture Organization of the United Nations (FAO), the world's area rice is about 157 million hectares. China and India are the major rice producing countries. During 1970 and 2019, global rice produced increased from 379 million tons to 514 million tons. Comments 3: Add a paragraph on GWAS and earlier studies in rice where it has been used to detect important QTLs. Response 3: Thank you for pointing this out. We agree with this comment. Therefore, we have added a paragraph as follows: Through large scale genomic data analysis, GWAS can identify genotypes associated with rice QTL such as yield, disease resistance and stress tolerance. Dong et al detected 30 QTL controlling rice tillering angle from 529 rice accessions, including TAC3, DWARF2 and TAC1. Chen et al detected 5 QTL for rice grain shape by measuring grain length and width in 289 different rice germplasms, including GS3, GIF1, GW5 and GSE9. Li et al identified the rice blast resistance gene bsr-d1 from rice Digu materials, which confers non-race-specific resistance to blast. Li et al identified a QTL qCTB4-1 that is significantly correlated with drought resistance during the rice boot stage from 121 rice materials collected from the Mini Core Collection. GWAS is performed using core accessions with strong polymorphisms in the target traits and has the advantage of studying quantitative traits in rice.
Comments 4: Line 52- Line 119- What was the basis of selection of 302 germplasm lines? Response 4: These 302 rice germplasms are selected from the 3K rice accessions. The frequency distribution histogram of the seed Protrusion percentage in 302 rice accessions samples shows a wide range of phenotypic variation and rich genetic variation. Population structure and phylogenetic analysis showed that these 302 rice accessions have population representativeness and are suitable for GWAS. Comments 5: Line 66- Specify standard field production techniques. Response 5: Normal field production was used for field management. Rice seeds were transferred to nursery beds for germination after soaking in water at 30°C for 48 hours. Seedlings of 20 days old were then transplanted to the paddy field. Comments 6: Line 74- Do you have image to show how this was done? Response 6: The following pictures show the way of my experiment. Comments 7: Where is the graph to show peak for K analysis for STRUCTURE? Response7: The following image is the graph to show peak for K analysis for STRUCTURE.
Figure 2. Population structure and phylogenetic analysis of 302 rice accessions. (a) Phylogenetic trees of 302 rice accessions. (b) PCA plots for the 302 rice accessions. (c) Cross validation error of K value. (d) ADMIXTURE analyses for k=9-13. Comments 8: Include the results for STRUCTURE analysis before moving to GWAS. Response 8: Thank you for pointing this out. We agree with this comment. Therefore, we have added a paragraph as follows: Genetic diversity and population structure of 302 rice germplasms were analyzed using 198712 molecular markers. According to the principle of maximum likelihood value, with ADMIXTURE 1.3.0 for population structure analysis, it was found that when K=11, CV error=0.70814, the smallest CV error value. Therefore, 302 varieties were divided into 11 subgroups (Figure 2c, d). The results of cluster analysis (Figure 2a) and principal component analysis (Figure 2b) are consistent with those of population structure analysis. |
|
|
|
|
|
Many grammatical or typographical errors have been revised. All the lines and pages indicated above are in the revised manuscript.
Thank you for the kind advice. Merry Christmas and happy New Year!
Sincerely yours, Xiaowen Ding
|

Round 2
Reviewer 1 Report
Comments and Suggestions for Authors
Dear Authors,
Thank you for addressing the concerns in this manuscript.
Reviewer 2 Report
Comments and Suggestions for Authors
Dear Authors,
Thank you for including the suggestions and answering the comments. The quality of the manuscript has improved tremendously.
Good luck!